# Mendelian randomization analysis reveals causal association of anthropometric measures on sepsis risk and mortality

**Chu-Yun Liu**[☯]**, Yu-Shen Yang**[☯]**, Meng-Qin Pei, He-fan He**[iD] *

Department of Anesthesiology, the Second Affiliated Hospital of Fujian Medical University, Quanzhou, Fujian Province, China

☯ These authors contributed equally to this work.
* 1017118837@fjmu.edu.cn

**Data Availability Statement:** The data underlying the results presented in the study are available from https://gwas.mrcieu.ac.uk/.

## Abstract

The objective of this study was to explore the potential causalities of fat mass, nonfat mass and height (henceforth, 'anthropometric measures') with sepsis risk and mortality. We conducted the Mendelian randomization (MR) investigation using genome-wide association study (GWAS) summary statistics of anthropometric measures, sepsis, and sepsis mortality. The GWAS summary data from the UK Biobank was used. Firstly, MR analysis was performed to estimate the causal effect of anthropometric measures on the risk of sepsis. The inverse-variance weighted (IVW) method was utilized as the primary analytical approach, together with weighted median-based method. Cochrane's Q test and MR-Egger intercept test were performed to assess heterogeneity and pleiotropy, respectively. Finally, we performed a series of sensitivity analyses to enhance the precision and veracity of our findings. The IVW method showed that genetically predicted weight-related measures were suggestively linked to an increased risk of sepsis. However, height displayed no causal association with sepsis risk and mortality. Furthermore, weight-related measures also displayed significant MR association with the sepsis mortality, except body nonfat mass and right leg nonfat mass. However, MVMR analysis indicated the observed effects for weight-related measures in the univariable MR analyses are more likely a bias caused by the interrelationship between anthropometric measures. According to the MR-Egger intercept assessment, our MR examination was not influenced by horizontal pleiotropy (all p>0.05). Moreover, the reliability of the estimated causal association was confirmed by the sensitivity analyses. In conclusion, these findings provided vital new knowledge on the role of anthropometric-related measures in the sepsis etiology.

## Introduction

Sepsis is a fatal organ dysfunction resulting from a dysregulated host response to infection [1]. It is reported that nearly 30,000,000 sepsis cases and 6,000,000 deaths globally occurred every year according to a latest systematic review [2]. More severely, the intensive care unit and

**Funding:** This work was supported by grants from the Natural Science Foundation of Fujian Province (2020J01227) and the Medical Innovation Science and Technology Project of Fujian Province (2020CXA047). The funder had vital role in study design, data collection and analysis, decision to publish, and preparation of the manuscript.

**Competing interests:** The authors have declared that no competing interests exist.

hospital mortality rates of sepsis are as high as 26 and 35%, respectively [3]. Therefore, identifying potential risk factors associated with sepsis is a pressing need in reducing its incidence and mortality rates.

Recently, obesity is indicated to be a risk factor for many diseases, including cardiovascular, inflammatory, metabolic and autoimmune diseases [4, 5]. Many literatures reported that excessive body fat can increase the risk of severe infectious disease by elevating the circulating proinflammatory proteins, leukocyte, neutrophil and monocyte counts [6, 7]. In sepsis, its' association with obesity has been revealed by some observational studies. For example, acceptable obesity was indicated to be a protective factor on the sepsis mortality, but morbid obesity and underweight were not [8]. Furthermore, a mendelian randomization (MR) study also demonstrated that there is a causal association between body mass index (BMI) and sepsis [9].

Nowadays, BMI (which is equal to weight [kg]/height [m]$^2$) is the most commonly used measure of obesity because its' ease of calculation and low cost [10]. However, it has received criticism recently for the following reasons [11]: 1) BMI falls short in its ability to distinguish between fat mass and non-fat mass (e.g., lean tissue mass), which might lead to the incorrect assessments of certain individuals. 2) BMI does not display the specific location of body fat which has been demonstrated to exert a vital role in forecasting the risk of several diseases.

Thus, increasing studies are taking an interest in thoroughly revealing the association between specific and biological components of BMI [including fat mass, nonfat mass and height (henceforth, 'anthropometric measures')] and disease. For example, in order to better understand the obesity–depression association, Speed et al. [12] performed a MR study of the relationship between anthropometric measures and depression, and they found that both height (short stature) and fat mass are causal risk factors for depression, while nonfat mass is not. In cardiometabolic diseases, Larsson et al. [13] also demonstrated that fat mass is causally associated with the risk of major cardiovascular events, while the effect of fat-free mass on cardiometabolic diseases seems to be neutral. These findings further confirmed the importance of fat mass in diseases via MR method, meanwhile denied the role of non-fat mass. However, the association of anthropometric measures with sepsis has not been revealed.

All of MR analysis were performed according to the following three key assumptions: (a) the multiple genetic variants included significantly involve in the exposure of interest; (b) no confounders affect the relationship between the exposure and outcome; (c) the genetic variants should influence the outcome only via their effect on exposure [14]. As a key epidemiological technique, MR analysis uses genetically instrumental variables (IVs) as proxies to explore the causality of exposure on outcome of interest, which overcomes these defects and bias caused by confounders, and other biases in observational studies [15]. Importantly, Pro. Minelli and coworkers [16] further demonstrated the feasibility of two sample MR method being used in one-sample MR analysis in 2021, which promoting the application and development of MR in medical field [17, 18].

Hence, this study was aimed at increasing the understanding of the obesity-sepsis association by evaluating the causal effects of anthropometric measures on sepsis risk and mortality via a MR study using results from large genome-wide association studies (GWAS).

## Methods

### Study data

We used an MR method to reveal the relationship between anthropometric measures and sepsis via employing summary statistics from GWAS. A total of 21 anthropometric measures were included and divided into three categories: general anthropometric measures (weight, BMI and height), fat mass (whole-body fat percentage, whole-body fat mass, and fat

percentage, fat mass for each of trunk, upper limbs, lower limbs), and nonfat mass (whole-body nonfat mass, nonfat mass for each of trunk, upper limbs, lower limbs). Genetic associations with these anthropometric measures can be found in **Supplementary Table 1 in S2 File**.

The summary statistics data for each anthropometric measure were obtained from Neale Lab (http://www.nealelab.is/uk-biobank/) who has performed the genome-wide association analysis for over 4200 phenotypes from the UK Biobank (http://www.ukbiobank.ac.uk/). With the aim at investigating the environmental and genetic contributors to disease, the UK Biobank recruits over 500,000 individuals of European ancestry and the average sample size for the anthropometric GWAS is 331,910. In addition, the genetic association estimations for SNPs linked to sepsis [IEU GWAS ID: ieu-b-4982; N = 431,365 (1,380 cases and 429,985 controls)] were obtained from the GWAS outcomes of the UK Biobank database. Specifically, sepsis cases were identified using 3rd international consensus on sepsis and septic shock issued by the American Medical Association in 2016 [1].

### Genetic instrumental variables selection

In this MR analysis, IVs were utilized to investigate the causality between anthropometric measures and sepsis risk. IVs (served as mediators between exposures and outcomes) were obtained from GWAS data for leading SNPs associated with anthropometric measures. The rigorous criteria were set to screen these IVs and all SNPs that showed strong association with the anthropometric measures at genome-wide significance ($P<5\times10^{-8}$) were selected. Then we harmonized these selected variants with sepsis data, excluding palindromic SNPs and clumping for linkage disequilibrium (LD, a window size greater than 10,000 kb, $r^2<0.001$) [19]. In addition, the mean F-statistic was used to assess the suitability of the selected SNPs for MR analysis. The SNPs with mean F statistics less than 10 were removed to avoid a weak instrument. The rigorous screen criteria and utilization of IVs insure the robustness of causal associations between anthropometric measures and the risk of sepsis.

### MR analysis

The common analytical methods were performed to examine the association between anthropometric measures and sepsis risk, including random-effects inverse-variance weighted (IVW) and weighted median approaches. As the primary MR analysis, IVW was used to account for heterogeneity among the causal evaluations from Wald ratio estimates [20, 21]. Furthermore, one other MR methods were employed for sensitivity analyses and further strengthen the findings. For sensitivity analyses, we used weighted median approaches. The weighted median requires that at least the 50% of the weight come from valid IVs [20]. Heterogeneity among individual genetic variant estimates was identified using Cochran's Q test and a P value below 0.05 provides potential evidence of heterogeneity. Additionally, the MR-Egger intercept test was performed to evaluate the possibility of horizontal pleiotropy and a p value of intercept<0.05 indicates a possible pleiotropy.

Importantly, multivariable MR (MVMR) was next conducted to evaluate the independent causal effects of significant anthropometric measures on sepsis outcomes. This method is an effective statistical approach, enabling us to estimate the direct effect of each phenotype on the outcome by incorporating SNPs with multiple phenotypes into the analysis [22]. The variants selected for MVMR analyses were screened based the following methods: i) the least absolute shrinkage and selection operator (LASSO) regression was used to remove highly correlated exposures in the presence of various exposures, enhancing the accuracy and reliability of the results [23]; ii) the variants with a Cochran's Q p value<0.05 were removed; iii) the remaining genetic instruments for each exposure with a significant effect on outcomes in the univariable

MR analyses were selected and then the MVMR analysis was then performed using TwoSampleMR R package. Because of the different screening criteria in the univariable and MVMR analyses, the number of IVs can be different.

### Interpretation of findings

If there was no evidence of horizontal pleiotropy, the IVW estimate was used as the most reliable indicator of the potential causal relationship; In the presence of evidence of horizontal pleiotropy and the weighted median estimates were performed as both proposed to correct for horizontal pleiotropic effects. We performed a Bonferroni correction with 42 tests to adjust for multiple testing in MR analysis, finally yielding a significance cutoff of $P \leq 0.001$ (0.05/42). Those exposures that were statistically significant ($P < 0.05$) before but not after correction ($P \leq 0.001$) were regarded as the suggestive correlation results.

In this study, all of MR analysis were performed by R software (version 4.2.0) and TwoSample MR package (version 0.5.6).

## Results

### IV selection for anthropometric measures

Details of the SNPs associated with 21 anthropometric measures, including the means standard deviations (SD), sample size, data source, population are shown in Supplementary Table 1 in S2 File. All SNPs obtained were strong IVs, and F values were all greater than 10, indicating that there was no bias caused by weak IVs in the study. The causal effects of each genetic variant on sepsis are shown via scatterplot, funnel plot and scatter plot, which were summarized in Supplementary Figures 1–14 in S1 File.

### Causal effect of general anthropometric measures with the risk of sepsis

The MR analysis conducted to estimate the causal effect of general anthropometric measures (BMI, weight and height) on the risk of sepsis is showed in Fig 1. In the primary IVW analyses, two general anthropometric measures (BMI and weight) showed strong MR association with the risk of sepsis. However, height exhibited no MR association with sepsis. Weighted median method also confirmed these associations between general anthropometric measures and sepsis risk.

### Causal effect of fat mass with the risk of sepsis

The MR analysis presented in Fig 2 estimated the causal effect of fat mass on the risk of sepsis. In the primary IVW analyses, the anthropometric measures of fat mass [including body fat percentage, body fat mass, arm fat percentage (right), arm fat percentage (left), leg fat percentage (right), leg fat percentage (left), trunk fat mass, arm fat mass (right), arm fat mass (left), leg fat mass (right), leg fat mass (left), trunk fat percentage] showed significantly strong MR association with the risk of sepsis (Fig 2). These results were demonstrated by Weighted median method, indicating that the anthropometric measures of fat mass had significantly causal MR association with the risk of risk with all p-values less than 0.05.

### Causal effect of nonfat mass with the risk of sepsis

Fig 3, which includes MR estimates obtained from various methods of assessing the causal effect of nonfat mass on the risk of sepsis, indicated that nonfat mass demonstrated the causal association with the risk of sepsis. In the primary IVW analyses, the anthropometric measures of nonfat mass [body nonfat mass, trunk nonfat mass, arm nonfat mass (right), arm nonfat mass (left), leg nonfat mass (right), leg nonfat mass (left)] exhibited strong MR association

| Exposures | Outcome | Methods | SNPs | | OR (95% CI) | adjust.P.value | Significance |
|---|---|---|---|---|---|---|---|
| Body_fat_percentage | sepsis | IVW | 249 | | 1.66(1.45 to 1.91) | 7.59e-13 | Yes |
| | | Weighted median | 249 | | 1.73(1.40 to 2.13) | 2.46e-07 | Yes |
| Body_fat_mass | sepsis | IVW | 276 | | 1.52(1.39 to 1.67) | 1.04e-19 | Yes |
| | | Weighted median | 276 | | 1.50(1.28 to 1.75) | 5.48e-07 | Yes |
| Arm_fat_percentage_(right) | sepsis | IVW | 231 | | 1.65(1.43 to 1.90) | 3.06e-12 | Yes |
| | | Weighted median | 231 | | 1.64(1.31 to 2.04) | 1.36e-05 | Yes |
| Arm_fat_percentage_(left) | sepsis | IVW | 248 | | 1.63(1.41 to 1.88) | 2.35e-11 | Yes |
| | | Weighted median | 248 | | 1.68(1.35 to 2.09) | 4.04e-06 | Yes |
| Leg_fat_percentage_(right) | sepsis | IVW | 244 | | 1.93(1.63 to 2.30) | 6.21e-14 | Yes |
| | | Weighted median | 244 | | 2.01(1.56 to 2.59) | 8.44e-08 | Yes |
| Leg_fat_percentage_(left) | sepsis | IVW | 245 | | 1.94(1.66 to 2.28) | 3.67e-16 | Yes |
| | | Weighted median | 245 | | 1.99(1.54 to 2.57) | 1.45e-07 | Yes |
| Trunk_fat_mass | sepsis | IVW | 277 | | 1.43(1.30 to 1.59) | 1.23e-12 | Yes |
| | | Weighted median | 277 | | 1.45(1.25 to 1.69) | 7.32e-07 | Yes |
| Arm_fat_mass_(right) | sepsis | IVW | 269 | | 1.56(1.42 to 1.70) | 1.53e-21 | Yes |
| | | Weighted median | 269 | | 1.51(1.29 to 1.76) | 1.63e-07 | Yes |
| Arm_fat_mass_(left) | sepsis | IVW | 266 | | 1.55(1.41 to 1.70) | 9.46e-21 | Yes |
| | | Weighted median | 266 | | 1.49(1.27 to 1.75) | 8.17e-07 | Yes |
| Leg_fat_mass_(right) | sepsis | IVW | 278 | | 1.72(1.53 to 1.92) | 2.25e-21 | Yes |
| | | Weighted median | 278 | | 1.68(1.40 to 2.02) | 3.86e-08 | Yes |
| Leg_fat_mass_(left) | sepsis | IVW | 278 | | 1.69(1.51 to 1.90) | 4.15e-20 | Yes |
| | | Weighted median | 278 | | 1.69(1.40 to 2.04) | 3.50e-08 | Yes |
| Trunk_fat_percentage | sepsis | IVW | 232 | | 1.47(1.30 to 1.66) | 5.97e-10 | Yes |
| | | Weighted median | 232 | | 1.55(1.31 to 1.83) | 4.53e-07 | Yes |

*adjust p value≤0.001 was considered statistically significant*

0   1   2        4
protective factor   risk factor

**Fig 1. Associations of general anthropometric measures with the risk of sepsis.** CI, confidence interval; IVW, inverse variance weighting; SNP: number of single-nucleotide polymorphisms that included in the analysis; OR, odds ratio.

with sepsis. The results from Weighted median method also supported the intimate association between nonfat mass and the risk of sepsis (all p < 0.05).

## Causal effect of anthropometric measures with the sepsis mortality

We estimated the causal effect of anthropometric measures on the sepsis mortality. In the primary IVW analyses, fat mass was suggestively associated with an increased risk of sepsis

| Exposures | Outcome | Methods | SNP | | OR (95% CI) | adjust.P.value | Significance |
|---|---|---|---|---|---|---|---|
| Body_nonfat_mass | sepsis | IVW | 394 | | 1.31(1.16 to 1.47) | 6.30e-06 | Yes |
| | | Weighted median | 394 | | 1.41(1.20 to 1.66) | 4.40e-05 | Yes |
| Trunk_nonfat_mass | sepsis | IVW | 392 | | 1.27(1.14 to 1.42) | 1.82e-05 | Yes |
| | | Weighted median | 392 | | 1.38(1.17 to 1.64) | 1.54e-04 | Yes |
| Arm_nonfat_mass_(right) | sepsis | IVW | 343 | | 1.35(1.19 to 1.53) | 4.16e-06 | Yes |
| | | Weighted median | 343 | | 1.50(1.24 to 1.81) | 2.43e-05 | Yes |
| Arm_nonfat_mass_(left) | sepsis | IVW | 348 | | 1.38(1.22 to 1.56) | 3.15e-07 | Yes |
| | | Weighted median | 348 | | 1.50(1.25 to 1.82) | 2.31e-05 | Yes |
| Leg_nonfat_mass_(right) | sepsis | IVW | 351 | | 1.47(1.30 to 1.67) | 1.07e-09 | Yes |
| | | Weighted median | 351 | | 1.53(1.28 to 1.84) | 5.53e-06 | Yes |
| Leg_nonfat_mass_(left) | sepsis | IVW | 350 | | 1.46(1.30 to 1.65) | 5.20e-10 | Yes |
| | | Weighted median | 350 | | 1.52(1.27 to 1.82) | 4.87e-06 | Yes |

*adjust p value≤0.001 was considered statistically significant*

0   1   2        4
protective factor   risk factor

**Fig 2. Associations of fat mass with the risk of sepsis.** CI, confidence interval; IVW, inverse variance weighting; SNP: number of single-nucleotide polymorphisms that included in the analysis; OR, odds ratio.

| Exposures | Outcome | Methods | SNP | | OR (95% CI) | adjust.P.value | Significance |
|---|---|---|---|---|---|---|---|
| BMI | sepsis | IVW | 298 | | 1.51(1.38 to 1.66) | 9.70e-18 | Yes |
| | | Weighted median | 298 | | 1.50(1.28 to 1.75) | 3.95e-07 | Yes |
| Weight | sepsis | IVW | 331 | | 1.44(1.31 to 1.59) | 9.74e-14 | Yes |
| | | Weighted median | 331 | | 1.44(1.24 to 1.67) | 2.03e-06 | Yes |
| Height | sepsis | IVW | 576 | | 1.05(0.99 to 1.12) | 1.32e-01 | No |
| | | Weighted median | 576 | | 1.07(0.96 to 1.18) | 2.03e-01 | No |

*adjust p value≤0.001 was considered statistically significant*

protective factor  risk factor

**Fig 3. Associations of nonfat mass with the risk of sepsis.** CI, confidence interval; IVX, inverse variance weighting; SNP: number of single-nucleotide polymorphisms that included in the analysis; OR, odds ratio.

mortality [BMI, weight, body fat percentage, body fat mass, arm fat percentage (right), arm fat percentage (left), leg fat percentage (right), leg fat percentage (left), trunk fat mass, arm fat mass (right), arm fat mass (left), leg fat mass (right), leg fat mass (left), trunk fat percentage] (Figs 4 and 5).

Some of general anthropometric and nonfat mass measures [including trunk nonfat mass, arm nonfat mass (right), arm nonfat mass (left), leg nonfat mass (left)] also showed significant MR association with sepsis mortality (Fig 6). However, height, body nonfat mass and leg nonfat mass (right) showed no MR association with sepsis mortality. Other method also indicated these association of anthropometric measures with the sepsis mortality.

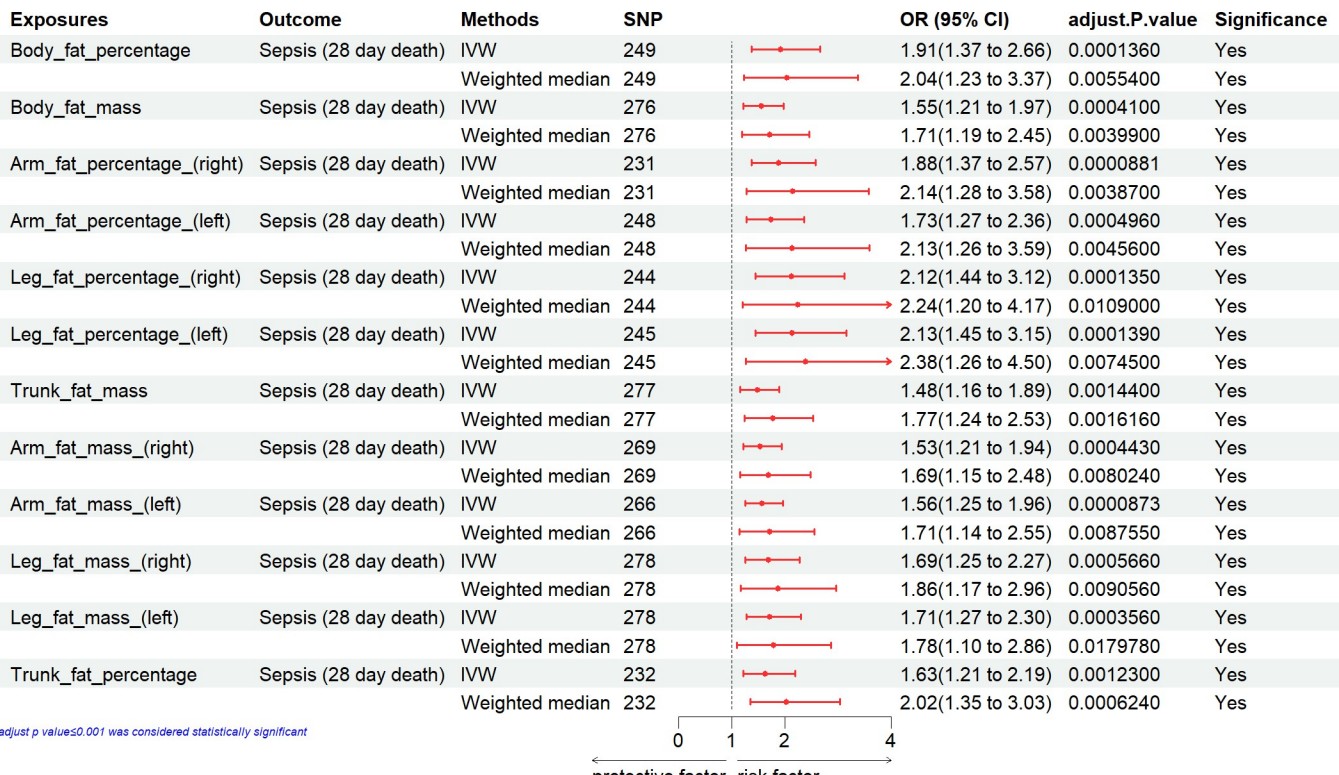

| Exposures | Outcome | Methods | SNP | | OR (95% CI) | adjust.P.value | Significance |
|---|---|---|---|---|---|---|---|
| Body_fat_percentage | Sepsis (28 day death) | IVW | 249 | | 1.91(1.37 to 2.66) | 0.0001360 | Yes |
| | | Weighted median | 249 | | 2.04(1.23 to 3.37) | 0.0055400 | Yes |
| Body_fat_mass | Sepsis (28 day death) | IVW | 276 | | 1.55(1.21 to 1.97) | 0.0004100 | Yes |
| | | Weighted median | 276 | | 1.71(1.19 to 2.45) | 0.0039900 | Yes |
| Arm_fat_percentage_(right) | Sepsis (28 day death) | IVW | 231 | | 1.88(1.37 to 2.57) | 0.0000881 | Yes |
| | | Weighted median | 231 | | 2.14(1.28 to 3.58) | 0.0038700 | Yes |
| Arm_fat_percentage_(left) | Sepsis (28 day death) | IVW | 248 | | 1.73(1.27 to 2.36) | 0.0004960 | Yes |
| | | Weighted median | 248 | | 2.13(1.26 to 3.59) | 0.0045600 | Yes |
| Leg_fat_percentage_(right) | Sepsis (28 day death) | IVW | 244 | | 2.12(1.44 to 3.12) | 0.0001350 | Yes |
| | | Weighted median | 244 | | 2.24(1.20 to 4.17) | 0.0109000 | Yes |
| Leg_fat_percentage_(left) | Sepsis (28 day death) | IVW | 245 | | 2.13(1.45 to 3.15) | 0.0001390 | Yes |
| | | Weighted median | 245 | | 2.38(1.26 to 4.50) | 0.0074500 | Yes |
| Trunk_fat_mass | Sepsis (28 day death) | IVW | 277 | | 1.48(1.16 to 1.89) | 0.0014400 | Yes |
| | | Weighted median | 277 | | 1.77(1.24 to 2.53) | 0.0016160 | Yes |
| Arm_fat_mass_(right) | Sepsis (28 day death) | IVW | 269 | | 1.53(1.21 to 1.94) | 0.0004430 | Yes |
| | | Weighted median | 269 | | 1.69(1.15 to 2.48) | 0.0080240 | Yes |
| Arm_fat_mass_(left) | Sepsis (28 day death) | IVW | 266 | | 1.56(1.25 to 1.96) | 0.0000873 | Yes |
| | | Weighted median | 266 | | 1.71(1.14 to 2.55) | 0.0087550 | Yes |
| Leg_fat_mass_(right) | Sepsis (28 day death) | IVW | 278 | | 1.69(1.25 to 2.27) | 0.0005660 | Yes |
| | | Weighted median | 278 | | 1.86(1.17 to 2.96) | 0.0090560 | Yes |
| Leg_fat_mass_(left) | Sepsis (28 day death) | IVW | 278 | | 1.71(1.27 to 2.30) | 0.0003560 | Yes |
| | | Weighted median | 278 | | 1.78(1.10 to 2.86) | 0.0179780 | Yes |
| Trunk_fat_percentage | Sepsis (28 day death) | IVW | 232 | | 1.63(1.21 to 2.19) | 0.0012300 | Yes |
| | | Weighted median | 232 | | 2.02(1.35 to 3.03) | 0.0006240 | Yes |

*adjust p value≤0.001 was considered statistically significant*

protective factor  risk factor

**Fig 4. Associations of general anthropometric measures with the mortality of sepsis.** CI, confidence interval; IVW, inverse variance weighting; SNP: number of single-nucleotide polymorphisms that included in the analysis; OR, odds ratio.

| Exposures | Outcome | Methods | SNPs | | OR (95% CI) | adjust.P.value | Significance |
|---|---|---|---|---|---|---|---|
| Body_nonfat_mass | Sepsis (28 day death) | IVW | 394 | | 1.28(1.00 to 1.65) | 0.054100 | No |
| | | Weighted median | 394 | | 1.52(1.03 to 2.24) | 0.034651 | Yes |
| Trunk_nonfat_mass | Sepsis (28 day death) | IVW | 392 | | 1.34(1.04 to 1.72) | 0.023600 | Yes |
| | | Weighted median | 392 | | 1.46(0.98 to 2.16) | 0.061192 | No |
| Arm_nonfat_mass_(right) | Sepsis (28 day death) | IVW | 343 | | 1.39(1.05 to 1.85) | 0.021900 | Yes |
| | | Weighted median | 343 | | 1.63(1.05 to 2.54) | 0.028450 | Yes |
| Arm_nonfat_mass_(left) | Sepsis (28 day death) | IVW | 348 | | 1.42(1.08 to 1.87) | 0.012900 | Yes |
| | | Weighted median | 348 | | 1.65(1.08 to 2.51) | 0.020497 | Yes |
| Leg_nonfat_mass_(right) | Sepsis (28 day death) | IVW | 351 | | 1.31(0.99 to 1.74) | 0.061600 | No |
| | | Weighted median | 351 | | 1.48(0.97 to 2.26) | 0.071152 | No |
| Leg_nonfat_mass_(left) | Sepsis (28 day death) | IVW | 350 | | 1.37(1.05 to 1.79) | 0.021100 | Yes |
| | | Weighted median | 350 | | 1.48(0.97 to 2.27) | 0.068533 | No |

*adjust p value≤0.001 was considered statistically significant*

protective factor  risk factor

**Fig 5. Associations of fat mass with the mortality of sepsis.** CI, confidence interval; IVW, inverse variance weighting; SNP: number of single-nucleotide polymorphisms that included in the analysis; OR, odds ratio.

## MVMR analyses

For the MVMR analyses, we constructed a model with anthropometric measures that retained an effect on sepsis risk and mortality in the univariable MR models. Supplementary Tables 2 and 3 in S2 File present the comprehensive estimated values and confidence intervals for the model included in the multivariable analysis. The positive results of MVMR, after accounting for various types of anthropometric measures, were as follows: arm fat percentage (left) and sepsis [OR (95% CI): 0.002(0.000–0.549), P = 0.03]; body fat mass and sepsis mortality [OR (95% CI): $1.39 \times 10^5$ ($2.745$–$7.023 \times 10^7$), P = 0.03]; leg fat percentage (left) and sepsis mortality [OR (95% CI): $2.061 \times 10^7$ ($1.580$–$2.69 \times 10^{14}$), P = 0.04]; leg fat mass (left) and sepsis mortality [OR (95% CI): 0.000(0.000–0.667), P < 0.05]; leg fat-free mass (left) and sepsis mortality [OR (95% CI): $1.825 \times 10^3$ ($5.231$–$6.367 \times 10^5$), P = 0.01]; arm fat percentage (left) and sepsis mortality [OR (95% CI): 0.000(0.000–0.050), P = 0.02] (Supplementary Table 2 and 3 in S2 File). Obviously, the wide 95%CI of OR indicated the multicollinearity among anthropometric measures. Thus, lasso regression analysis was used to remove highly correlated exposures. In lasso regression model, four anthropometric measures (BMI, weight, leg fat mass (right) and arm fat-free mass (left)) were selected for sepsis risk evaluation and three anthropometric measures (body fat mass, leg fat mass (right) and leg fat-free mass (left)) for sepsis mortality evaluation. The results revealed that compared with aforementioned estimates, no causal association was observed between these selected anthropometric measures and sepsis (Supplementary Fig 15 in S1 File). Thus, the observed effects for anthropometric measures in the univariable MR analyses are more likely operating through the pathways of other anthropometric measures.

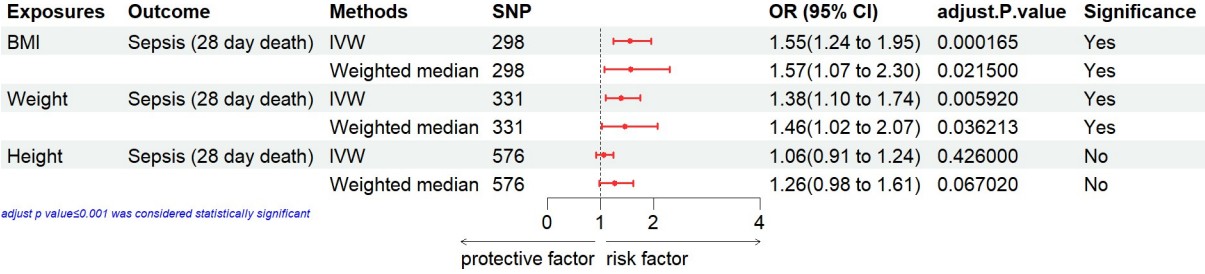

| Exposures | Outcome | Methods | SNP | | OR (95% CI) | adjust.P.value | Significance |
|---|---|---|---|---|---|---|---|
| BMI | Sepsis (28 day death) | IVW | 298 | | 1.55(1.24 to 1.95) | 0.000165 | Yes |
| | | Weighted median | 298 | | 1.57(1.07 to 2.30) | 0.021500 | Yes |
| Weight | Sepsis (28 day death) | IVW | 331 | | 1.38(1.10 to 1.74) | 0.005920 | Yes |
| | | Weighted median | 331 | | 1.46(1.02 to 2.07) | 0.036213 | Yes |
| Height | Sepsis (28 day death) | IVW | 576 | | 1.06(0.91 to 1.24) | 0.426000 | No |
| | | Weighted median | 576 | | 1.26(0.98 to 1.61) | 0.067020 | No |

*adjust p value≤0.001 was considered statistically significant*

protective factor  risk factor

**Fig 6. Associations of nonfat mass with the mortality of sepsis.** CI, confidence interval; IVX, inverse variance weighting; No. of SNPs: number of single-nucleotide polymorphisms that included in the analysis; OR, odds ratio.

## Heterogeneity, horizontal pleiotropy, and sensitivity analysis

Table 1 displays the findings from Cochrane's Q test and pleiotropy test. Specifically, our analysis suggested that potential heterogeneity was observed in the MR analyses of some anthropometric measures and sepsis. The significant heterogeneity will affect the results of Mendelian

Table 1. Robustness of anthropometric measures-sepsis.

| Exposures | Outcomes | Cochrane's Q test | | Pleiotropy assessment | | |
|---|---|---|---|---|---|---|
| | | Q | P value | Egger intercept | se | P value |
| BMI | Sepsis | 362.307 | 0.006 | 0.002 | 0.003 | 0.523 |
| | Sepsis mortality | 358.305 | 0.008 | 0.001 | 0.007 | 0.9 |
| Body fat percentage | Sepsis | 302.966 | 0.01 | 0.004 | 0.004 | 0.261 |
| | Sepsis mortality | 290.375 | 0.033 | -0.001 | 0.009 | 0.922 |
| Body fat mass | Sepsis | 305.746 | 0.098 | 0.001 | 0.003 | 0.74 |
| | Sepsis mortality | 328.207 | 0.015 | -0.004 | 0.008 | 0.596 |
| Arm fat percentage (right) | Sepsis | 274.215 | 0.024 | 0.003 | 0.004 | 0.384 |
| | Sepsis mortality | 259.113 | 0.091 | 0.002 | 0.009 | 0.794 |
| Arm fat percentage (left) | Sepsis | 312.863 | 0.003 | 0.003 | 0.004 | 0.33 |
| | Sepsis mortality | 278.324 | 0.083 | 0 | 0.008 | 0.952 |
| Leg fat percentage (right) | Sepsis | 288.346 | 0.024 | 0.003 | 0.004 | 0.446 |
| | Sepsis mortality | 270.662 | 0.107 | -0.002 | 0.01 | 0.862 |
| Leg fat percentage (left) | Sepsis | 276.41 | 0.075 | 0.002 | 0.004 | 0.611 |
| | Sepsis mortality | 269.129 | 0.129 | -0.01 | 0.009 | 0.281 |
| Trunk fat mass | Sepsis | 351.539 | 0.001 | 0.003 | 0.003 | 0.408 |
| | Sepsis mortality | 349.911 | 0.002 | -0.006 | 0.008 | 0.486 |
| Arm fat mass (right) | Sepsis | 279.664 | 0.3 | 0.004 | 0.003 | 0.168 |
| | Sepsis mortality | 307.293 | 0.05 | -0.004 | 0.007 | 0.574 |
| Arm fat mass (left) | Sepsis | 293.583 | 0.11 | 0.002 | 0.003 | 0.576 |
| | Sepsis mortality | 303.201 | 0.053 | -0.005 | 0.007 | 0.513 |
| Leg fat mass (right) | Sepsis | 293.043 | 0.243 | 0.001 | 0.003 | 0.701 |
| | Sepsis mortality | 330.126 | 0.016 | -0.001 | 0.008 | 0.88 |
| Leg fat mass (left) | Sepsis | 303.79 | 0.129 | 0.001 | 0.003 | 0.839 |
| | Sepsis mortality | 318.971 | 0.042 | -0.001 | 0.007 | 0.937 |
| Weight | Sepsis | 391.061 | 0.012 | 0.002 | 0.003 | 0.407 |
| | Sepsis mortality | 374.215 | 0.047 | 0.003 | 0.006 | 0.609 |
| Height | Sepsis | 610.154 | 0.15 | 0 | 0.002 | 0.939 |
| | Sepsis mortality | 561.504 | 0.649 | -0.005 | 0.004 | 0.134 |
| Body nonfat mass | Sepsis | 485.565 | 0.001 | -0.001 | 0.002 | 0.563 |
| | Sepsis mortality | 414.397 | 0.22 | -0.006 | 0.005 | 0.252 |
| Trunk fat percentage | Sepsis | 294.336 | 0.003 | 0.007 | 0.004 | 0.086 |
| | Sepsis mortality | 287.654 | 0.007 | 0.002 | 0.01 | 0.808 |
| Trunk nonfat mass | Sepsis | 451.883 | 0.018 | 0 | 0.002 | 0.969 |
| | Sepsis mortality | 374.672 | 0.715 | -0.008 | 0.005 | 0.111 |
| Arm nonfat mass (right) | Sepsis | 411.835 | 0.006 | 0.001 | 0.003 | 0.668 |
| | Sepsis mortality | 355.77 | 0.293 | -0.007 | 0.006 | 0.252 |
| Arm nonfat mass (left) | Sepsis | 408.674 | 0.013 | 0.001 | 0.002 | 0.732 |
| | Sepsis mortality | 385.029 | 0.078 | -0.004 | 0.006 | 0.531 |
| Leg nonfat mass (right) | Sepsis | 457.005 | 0 | 0.001 | 0.003 | 0.572 |
| | Sepsis mortality | 396.289 | 0.044 | -0.006 | 0.006 | 0.281 |
| Leg nonfat mass (left) | Sepsis | 423.896 | 0.004 | 0 | 0.002 | 0.986 |
| | Sepsis mortality | 389.451 | 0.067 | -0.008 | 0.006 | 0.176 |

randomization analysis. Thus, in this study, if there is no heterogeneity (P-value>0.05), the fixed-effect IVW model was applied; otherwise (P-value≤0.05), random-effects IVW model was applied. If there is statistical evidence for a causal effect in a random-effects analysis, this means that there is consistent evidence that the genetic variants support a causal effect of the exposure on the outcome even accounting for heterogeneity in the variant-specific causal estimates. Nevertheless, the MR-Egger intercept tests did not reveal any evidence of horizontal pleiotropy during the MR analyses, as all p values were above 0.05. Taken together, all of the findings suggest that the association between genetic predisposition to anthropometric measures and sepsis was significantly impacted by any individual SNP.

## Discussion

This study was aimed at investigating the causal role of different anthropometric measures on sepsis to gain a better understanding of the impact of fat and non-fat mass on sepsis risk. Our findings indicated that weight-related measures are closely related to sepsis risk and mortality, but height are not. Our study provides evidence that BMI shows positive association with sepsis.

Due to the inability of BMI to discriminate the specific type and location of body fat, this study utilized different anthropometric measures that enable to reflect the stored compartments of fat and differentiate between fat and non-fat tissues. Our main finding is that people with greater weight are more susceptible to sepsis; however, height does not display causal association with sepsis. These findings demonstrated that the aforementioned BMI-sepsis causality is probably driven by weight factor. In addition, we also attempted to explore whether the location influenced the strength of the causal association between weight-related measures and sepsis. The results did not indicate an overwhelming winner that has significantly stronger influence on the risk of sepsis than other measures after correcting for multiple comparisons.

In fact, the association between obesity and sepsis has been reported by many experimental and clinical studies. It was reported that obesity can induce the inflammatory response in sepsis patients through promoting the overproduction of proinflammatory factors (i.e., tumor necrosis factor). Then the initiation of a cascade of inflammation facilitates the upregulation of systemic immunity (i.e., metabolic syndrome and insulin resistance) and thus increase the risk of infection, organ failure and death in patients [24]. Furthermore, worsened oxidative stress in obese patients with sepsis is regarded as another explanation. An animal experiment using sepsis rat model demonstrated that obese rats with sepsis display oxidative stress mainly in the lung and liver reflected by an oxidative damage to lipids and proteins and an imbalance of superoxide dismutase and catalase levels [25].

The BMI has always been used as a surrogate marker for classifying obesity and is associated with risk factors for sepsis. In a Population-Based Cohort Study, the authors indicated that obesity (BMI ≥40) is independently related to future sepsis events, but waist circumference, by contrast, presents better predictive property for future sepsis risk than BMI [26]. Indeed, the use of BMI to identify obese persons has been criticized. Currently, a new term "obesity paradox" is now popular in academic world. The reason is that high BMI is demonstrated to contribute to worse outcomes for critically ill patients in early works [27, 28], while a relationship with improved outcomes with an raised BMI was found in some more recent work [29–31]. The underlying mechanism of this phenomenon remains unclear and understanding the genetic role of sepsis is perhaps the key to unravel the mysterious relationship between obesity and outcomes in patients with sepsis. Therefore, we used MR studies to evaluate the causal effect estimates between exposure factors and outcome variables by using genetic IV. Actually, the causal association between BMI and sepsis risk and mortality has been reported by other

literatures. For example, a latest MR study indicated the positive association of BMI with sepsis mortality at 28 days [32], and another MR demonstrated their causal link of high BMI and an increased risk of sepsis [33]. Our results from univariable MR analysis supported above findings. However, these MR analyses failed to control for pleiotropic pathways resulted from the multicollinearity among anthropometric measures. Thus, we further performed MVMR analyses to assess the independent causal effects of BMI on sepsis risk utilizing lasso regression method. Our results demonstrated that the direct estimates for the BMI were significantly attenuated, causing a wider 95% CI that overlapped null. The same phenomenon was also seen in other anthropometric measures displaying significant causality with sepsis in univariable MR analysis. Hence, the observed effects for BMI in the univariable MR analyses are more likely a bias caused by the interrelationship between anthropometric measures.

It is worth mentioning that the association of lean mass (namely nonfat mass, mainly including muscle and bone) and body fat change with sepsis has also been revealed in recent literature. Several literatures have demonstrated that septic humans and animals are in great demand for supraphysiological energy supplies, but they are usually unwilling to eat or can't eat. Such negative energy balance leads to the amounts of fat tissue, cellular glycogen, and muscle mass decline rapidly, in order to provide enough energy for body metabolism [34]. However, these high-energy metabolites [including free fatty acids, glycerol, lactate, and gluconeogenic amino acids] can't be transformed into useful molecules such as glucose and ketones to provide and release energy, due to the downregulation of GR and PPARα. Thus, instead of supplying energy for body, the decomposition of lean mass and fat mass leads to the accumulation and toxicity of metabolites, further causing the aggravation of disease severity and lethality in septic patients and animals. For example, Lee et al revealed that muscle mass loss/depletion was associated with increased sepsis mortality [35]. Importantly, some studies further suggested that obese septic mice show a unique metabolic profile that is characterized by enhanced lipolysis when compared to lean septic mice [36]. These finding indicated that there is a positive correlation of sepsis severity with body mass (including lean mass and fat mass). However, it remains unclear whether the demonstrated associations are causal. In order to reveal this mystery, we performed this MR analysis and found that genetically mediated nonfat mass and fat mass are causally associated with an increased risk of sepsis via univariable MR analyses.

Our work also had some potential limitations. Firstly, only European populations were included. However, different cohorts of patients have different genetic profiles due to the difference of ethnic diversity, risk factors, and epigenetic profiles. Thus, the reliability of the causal relationships from our analysis should be demonstrated in other non-European descent. Secondly, body composition is affected by sex and aging process. For example, generally, men have more muscle mass and women have greater fat mass. In the process of aging, total body fat mass gradually increases, and lean mass and bone density proportionally decreases, which are independent of general and physiological changes in BMI. The causal link between anthropometric measures and sepsis in men and women of different ages could be different due to the changes of body composition. Thus, additional studies should be performed to determine which anthropometric measures could strongly assess the risk of sepsis among obese men and women of different ages. Thirdly, due to the lack of variants with strong association with sepsis ($p < 5 \times 10^{-8}$), we have not constructed a possible bidirectional causal relationship between sepsis and anthropometric measures. This could impede the explanation about whether sepsis in turn causes higher anthropometric measures.

In conclusion, the current work firstly provides evidence that weight-related measures have a significangt association on sepsis risk and mortality via univariable MR analyses, but MVMR analysis indicated the observed effects for weight-related measures in the univariable MR

analyses are more likely a bias caused by the interrelationship between anthropometric measures. This probably explain the reason why "obesity paradox" occur in sepsis: namely previous literatures only paid attention to the association between BMI and sepsis and did not consider the collinear effect from other anthropometric measures. Thus, these findings represent vital new knowledge on the role of anthropometric-related measures in the sepsis etiology.

## Supporting information

**S1 Checklist. Human participants research checklist.**
(DOCX)

**S2 Checklist. STROBE-MR checklist of recommended items to address in reports of Mendelian randomization studies.**
(DOCX)

**S1 File.**
(DOCX)

**S2 File.**
(XLSX)

## Author Contributions

**Conceptualization:** Chu-Yun Liu, He-fan He.

**Data curation:** Chu-Yun Liu, Yu-Shen Yang.

**Formal analysis:** Chu-Yun Liu, Yu-Shen Yang, He-fan He.

**Funding acquisition:** He-fan He.

**Investigation:** Meng-Qin Pei.

**Methodology:** Chu-Yun Liu, Yu-Shen Yang, Meng-Qin Pei.

**Project administration:** He-fan He.

**Software:** Chu-Yun Liu, Yu-Shen Yang, Meng-Qin Pei.

**Supervision:** He-fan He.

**Validation:** He-fan He.

**Writing – original draft:** Chu-Yun Liu.

**Writing – review & editing:** He-fan He.

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
