## [Decision Letter · Decision Letter 0]

6 Dec 2023

PONE-D-23-24676Mendelian randomization analysis reveals causal association of anthropometric measures on sepsis risk and mortalityPLOS ONE

Dear Dr. He,

Thank you for submitting your manuscript to PLOS ONE. After careful consideration, we feel that it has merit but does not fully meet PLOS ONE’s publication criteria as it currently stands. Therefore, we invite you to submit a revised version of the manuscript that addresses the points raised during the review process.

**The manuscript addresses a topic of potential interest. However, the study exhibits significant pitfalls that need addressing to fortify the validity of conclusions. To highlight a few: i) provide explanations for potential factors leading to a noteworthy Cochrane's Q test and an inconsequential MR-Egger intercept test; ii) tackle potential bias in your methodology or discussion such as the handling of the interrelationship between anthropometric measures, which could introduce bias; iii) furnish more methodological details, including the methods employed for LD clumping and MR, as well as the correction applied for multiple tests; iv) display MR results through scatter plots or other visuals, and include leave-one-out forest plots for the main analyses at a minimum; v) explore the potential association between non-fat mass and sepsis; vi) adhere to STROBE-MR guidelines.**

We look forward to receiving your revised manuscript.

Kind regards,

Giuseppe Remuzzi

Academic Editor

PLOS ONE

Journal Requirements:

3. PLOS requires an ORCID iD for the corresponding author in Editorial Manager on papers submitted after December 6th, 2016. Please ensure that you have an ORCID iD and that it is validated in Editorial Manager. To do this, go to ‘Update my Information’ (in the upper left-hand corner of the main menu), and click on the Fetch/Validate link next to the ORCID field. This will take you to the ORCID site and allow you to create a new iD or authenticate a pre-existing iD in Editorial Manager. Please see the following video for instructions on linking an ORCID iD to your Editorial Manager account: https://www.youtube.com/watch?v=_xcclfuvtxQ".

4. Please amend your authorship list in your manuscript file to include author Meng-Qin Pei.

Reviewers' comments:

Reviewer's Responses to Questions

**Comments to the Author**

1. Is the manuscript technically sound, and do the data support the conclusions?

Reviewer #1: Partly

Reviewer #2: Partly

2. Has the statistical analysis been performed appropriately and rigorously? 

Reviewer #1: Yes

Reviewer #2: N/A

3. Have the authors made all data underlying the findings in their manuscript fully available?

Reviewer #1: Yes

Reviewer #2: Yes

4. Is the manuscript presented in an intelligible fashion and written in standard English?

Reviewer #1: No

Reviewer #2: Yes

5. Review Comments to the Author

Reviewer #1: I would like to commend you for your work on investigating the causal relationship between anthropometric measures and sepsis risk. It addresses a significant question with the potential for important clinical implications. However, after a thorough review, several aspects of the manuscript require further clarification and expansion.

Introduction/Discussion: The introduction and discussion sections of your paper seem somewhat vague. The depth of discussion could be improved to better align with the results. While the introduction primarily discusses the contrasting findings of BMI and obesity related to sepsis risk, the main results appear to focus on fat mass parameters. It would be beneficial if these aspects were more thoroughly addressed and integrated into the research aims and background.

Results: There appears to be an error in the reporting of results on lines 177-178. The statement reads: "Weighted median method also supported the intimated association between nonfat mass and the risk of sepsis (all p>0.05)". Based on the Mendelian Randomization theory and the figures presented in Fig 3, the p-values estimated by the weighted median method should all be less than 0.05.

Interpretation of Cochrane's Q tests and MR-Egger Intercept tests: The interpretation of these tests seems to lack sufficient discussion. Kindly provide explanations for potential reasons that could lead to a significant Cochrane's Q test and an insignificant MR-Egger intercept test.

Interrelationship between Anthropometric Measures: It would be helpful if you could discuss how you addressed the interrelationship between anthropometric measures, which could introduce bias. If the exposures are highly correlated (i.e., collinear), it becomes challenging to isolate the causal effects of individual exposures. This is due to the possibility of genetic variants used as instrumental variables (IVs) for one exposure also being associated with other exposures—a problem further compounded in cases of multicollinearity.

Same Population for Exposure and Outcome Data: Using the same population for all exposure and outcome data can introduce sample overlap bias. Please address this potential bias in your methodology or discussion.

In conclusion, your research addresses an important topic, and with some additional work, it could provide valuable insights into the study of sepsis risk. I look forward to your response and the subsequent version of your manuscript.

Best Regards,

Reviewer #2: The authors performed a mendelian randomization (MR) study to investigate the link between anthropometric traits such as weight, BMI, height, etc., and sepsis and sepsis death. The manuscript lacks a detailed description of the methods used and plots that are normally included in MR studies (see comments below).

This is basically a one-sample MR since both the GWAS on the exposure and that on the outcome came from the same cohort (UK biobank). As this approach possess some limitations, these should be discussed in the text. Especially because two-samples MR studies on BMI and sepsis already exists.

- The methods used for LD clumping and MR should be described and the relevant references and software cited.

- Methods: "The weighted median yielded consistent estimates even when over 50% of genetic SNPs have horizontal pleiotropy" This is a little ambiguous, I would rephrase it with something like "The weighted median requires that at least the 50% of the weight come from valid IVs".

- The last sentence of the methods mentions a correction for multiple tests. However, it is not clear how and where it was applied.

- The results contain long lists of the estimated OR, CI and p-value that are already reported in the tables and that make the text hard to read

- Results line 178: "(all p > 0.05 )" should probably be "(all p < 0.05)".

- All tables are reported as figures, which make them difficult to read. Moreover, a table with the markers used for the analyses could be reported in supplementary tables.

- It is common practice to display the results of MRs by a scatterplot or other plots. Also, consider to include leave-one-out forest plots, at least for the main analyses.

- In the discussion, the potential role of obesity and BMI on sepsis is discussed. As other studies already showed an association between BMI and sepsis by two-sample MR, the authors could elaborate on the association between non-fat mass and sepsis.

- Although I am not an expert of the field, I think it might be worth to discuss the possible effect of confounders on the relationship between BMI and mortality due to sepsis since BMI has been linked to mortality in many ways.

- Compliance with Ethics Guidelines: the authors state that the manuscript follows the STROBE-MR guide lines. I disagree on this point. The STROBE-MR paper contains a detailed table describing how to report the results of MR based studies. In my opinion, the manuscript is not structured according to such guidelines.

6. PLOS authors have the option to publish the peer review history of their article (what does this mean?). If published, this will include your full peer review and any attached files.

Reviewer #1: No

Reviewer #2: **Yes: **Matteo Breno

---

## [Author Response · Author response to Decision Letter 0]

16 Jan 2024

Reviewers' Comments:

General response: We thank the reviewers for their time and insightful comments. We have carefully considered these comments and substantially revised the manuscript by thoroughly researching the field reviewed and clarifying the content of this manuscript to make it more comprehensive. Below, we address each of the reviewers’ questions.

Reviewer 1

I would like to commend you for your work on investigating the causal relationship between anthropometric measures and sepsis risk. It addresses a significant question with the potential for important clinical implications. However, after a thorough review, several aspects of the manuscript require further clarification and expansion.

Our response: We appreciate the reviewer’s critical and constructive comments. It is our honour to get your help to revise our manuscript. Your comments give us valuable assistance in improving our manuscript. We have carefully revised our manuscript accordingly. After adopting your suggestions, we believe a significant improvement can be seen in our revised manuscript. The following are detailed responses to your recommendations.

1. Introduction/Discussion: The introduction and discussion sections of your paper seem somewhat vague. The depth of discussion could be improved to better align with the results. While the introduction primarily discusses the contrasting findings of BMI and obesity related to sepsis risk, the main results appear to focus on fat mass parameters. It would be beneficial if these aspects were more thoroughly addressed and integrated into the research aims and background.

Our response: Thank you for your feedback and comments regarding the vagueness of the introduction and discussion sections. We are very grateful that the reviewer read our article carefully. As we all know, currently, BMI, the most common fat mass parameters, is commonly used to identify obese persons due to its ease of calculation and cost-effectiveness. However, its use has been repeatedly criticized for its shortcomings. For example, BMI does not distinguish between fat mass and nonfat mass, and dose not capture body fat location, but different types of tissue mass and different fat location have been demonstrated to play different role in the development of disease. As precision medicine continue developing, it is necessary to investigate and determine the specific effect of different types of tissue mass and different fat location on disease. Thus, this study was aimed at increasing the understanding of the obesity-sepsis association by evaluating the causal effects of fat mass, nonfat mass and height (henceforth, ‘anthropometric measures’) on sepsis risk and mortality via a MR study using results from large genome-wide association studies (GWAS). On the other hand, the tissue mass of different types and locations can be reflected by detailed fat mass parameters. That’s why the main results in this study appear to focus on fat mass parameters. Undoubtedly, your suggestion is of great help to improve the quality and preciseness of this manuscript. Hence, we have made some revisions, which was showed in red. 

2. Results: There appears to be an error in the reporting of results on lines 177-178. The statement reads: "Weighted median method also supported the intimated association between nonfat mass and the risk of sepsis (all p>0.05)". Based on the Mendelian Randomization theory and the figures presented in Fig 3, the p-values estimated by the weighted median method should all be less than 0.05.

Our response: It’s our negligence for this mistake. We have revised this error. Thanks very much for your careful review again. Even so, we are still really sorry for our cursoriness. Please see Page 8-9, Lines 172-174.

(Lines 172-174): 

The results from Weighted median method also supported the intimate association between nonfat mass and the risk of sepsis (all p < 0.05).

3. Interpretation of Cochrane's Q tests and MR-Egger Intercept tests: The interpretation of these tests seems to lack sufficient discussion. Kindly provide explanations for potential reasons that could lead to a significant Cochrane's Q test and an insignificant MR-Egger intercept test.

Our response: We appreciate your good suggestion on improving the accuracy of the expression. The potential cause of the heterogeneity is probably involved in instrumental variables from different analysis platforms, experiments, populations, etc., The significant heterogeneity will affect the results of Mendelian randomization analysis. Thus, we usually need to evaluate the heterogeneity of genetic variant using Cochran’s Q test at first. If there is no excess heterogeneity in the variant-specific causal estimates, then the random-effects and fixed-effect results will be identical. So, there is no loss of precision. If there is excess heterogeneity in the variant specific causal estimates, then the fixed-effect estimate is overly precise, and this heterogeneity should be accounted for. If there is statistical evidence for a causal effect in a random-effects analysis, this means that there is consistent evidence that the genetic variants support a causal effect of the exposure on the outcome even accounting for heterogeneity in the variant-specific causal estimates. Thus, in this study, if there is heterogeneity (P-value≤0.05), random-effects IVW models are applied; otherwise (P-value>0.05), the fixed-effect IVW model is applied. 

On the other hand, in order to evaluate whether instrumental variables influence outcomes through factors other than exposure, pleiotropic analysis was performed using MR-Egger intercept test in this study. The results did not reveal any evidence of horizontal pleiotropy during the MR analyses, as all p values were above 0.05.

However, it’s our negligence for the insufficient discussion of the interpretation of these tests. Thus, we supplemented it in our manuscript. Please see Page 12, Lines 229-241.

(Lines 229-241):

Table 1 displays the findings from Cochrane’s Q test and pleiotropy test. Specifically, our analysis suggested that potential heterogeneity was observed in the MR analyses of some anthropometric measures and sepsis. The significant heterogeneity will affect the results of Mendelian randomization analysis. Thus, in this study, if there is no heterogeneity (P-value>0.05), the fixed-effect IVW model was applied; otherwise (P-value≤0.05), random-effects IVW model was applied. If there is statistical evidence for a causal effect in a random-effects analysis, this means that there is consistent evidence that the genetic variants support a causal effect of the exposure on the outcome even accounting for heterogeneity in the variant-specific causal estimates. Nevertheless, the MR-Egger intercept tests did not reveal any evidence of horizontal pleiotropy during the MR analyses, as all p values were above 0.05. Taken together, all of the findings suggest that the association between genetic predisposition to anthropometric measures and sepsis was significantly impacted by any individual SNP.

4. Interrelationship between Anthropometric Measures: It would be helpful if you could discuss how you addressed the interrelationship between anthropometric measures, which could introduce bias. If the exposures are highly correlated (i.e., collinear), it becomes challenging to isolate the causal effects of individual exposures. This is due to the possibility of genetic variants used as instrumental variables (IVs) for one exposure also being associated with other exposures—a problem further compounded in cases of multicollinearity.

Our response: Thanks very much for your constructive suggestions. You are right, if the exposures are highly correlated (i.e., collinear), it becomes challenging to isolate the causal effects of individual exposures. Thus, we performed multivariable MR to evaluate the independent causal effects of relevant anthropometric measures on sepsis outcomes. However, the positive results of MVMR showed very wide 95%CI of OR, which indicating the multicollinearity among anthropometric measures. Thus, lasso regression analysis was used to remove highly correlated exposures. In lasso regression model, four anthropometric measures (BMI, weight, leg fat mass (right) and arm fat-free mass (left)) were selected for sepsis risk evaluation and three anthropometric measures (body fat mass, leg fat mass (right) and leg fat-free mass (left)) for sepsis mortality evaluation. The results revealed that compared with aforementioned estimates, no causal association was observed between these selected anthropometric measures and sepsis (Supplementary Fig 15). Thus, the observed effects for weight-related measures in the univariable MR analyses are more likely a bias caused by the interrelationship between anthropometric measures. These findings probably provide a novel understanding for the reason why “obesity paradox” occur in sepsis: namely previous literatures only paid attention to the association between BMI and sepsis and did not consider the collinear effect from other anthropometric measures. Please see Page 5, Lines 115-121; Page 11-12, Lines 206-227; Page 15, Lines 293-300.

(Lines 115-121):

Importantly, multivariable MR (MVMR) was next conducted to evaluate the independent causal effects of relevant anthropometric measures on sepsis outcomes. This method is an effective statistical approach, enabling us to estimate the direct effect of each phenotype on the outcome by incorporating SNPs with multiple phenotypes into the analysis1. Here, the least absolute shrinkage and selection operator (LASSO) regression was used to remove highly correlated exposures in the presence of various exposures, enhancing the accuracy and reliability of the results2.

(Lines 204-226):

For the MVMR analyses, we constructed a model with anthropometric measures that retained an effect on sepsis risk and mortality in the univariable MR models. Supplementary table 2 and 3 present the comprehensive estimated values and confidence intervals for the model included in the multivariable analysis. The positive results of MVMR, after accounting for various types of anthropometric measures, were as follows: arm fat percentage (left) and sepsis [OR (95% CI): 0.002(0.000–0.549), P = 0.03]; body fat mass and sepsis mortality [OR (95% CI): 1.39 x10^5 (2.745–7.023 x10^7), P = 0.03]; leg fat percentage (left) and sepsis mortality [OR (95% CI): 2.061x10^7 (1.580–2.69x10^14), P = 0.04]; leg fat mass (left) and sepsis mortality [OR (95% CI): 0.000(0.000–0.667), P < 0.05]; leg fat-free mass (left) and sepsis mortality [OR (95% CI): 1.825 x10^3 (5.231–6.367 x10^5), P = 0.01]; arm fat percentage (left) and sepsis mortality [OR (95% CI): 0.000(0.000–0.050), P = 0.02] (Supplementary table 2-3). Obviously, the wide 95%CI of OR indicated the multicollinearity among anthropometric measures. Thus, lasso regression analysis was used to remove highly correlated exposures. In lasso regression model, four anthropometric measures (BMI, weight, leg fat mass (right) and arm fat-free mass (left)) were selected for sepsis risk evaluation and three anthropometric measures (body fat mass, leg fat mass (right) and leg fat-free mass (left)) for sepsis mortality evaluation. The results revealed that compared with aforementioned estimates, no causal association was observed between these selected anthropometric measures and sepsis (Supplementary Fig 15). Thus, the observed effects for anthropometric measures in the univariable MR analyses are more likely operating through the pathways of other anthropometric measures.

(Lines 293-300):

Thus, we further performed MVMR analyses to assess the independent causal effects of BMI on sepsis risk utilizing lasso regression method. Our results demonstrated that the direct estimates for the BMI were significantly attenuated, causing a wider 95% CI that overlapped null. The same phenomenon was also seen in other anthropometric measures displaying significant causality with sepsis in univariable MR analysis. Hence, the observed effects for BMI in the univariable MR analyses are more likely a bias caused by the interrelationship between anthropometric measures.

5. Same Population for Exposure and Outcome Data: Using the same population for all exposure and outcome data can introduce sample overlap bias. Please address this potential bias in your methodology or discussion.

Our response: We are very grateful that the reviewer read our article carefully and has provided this idea. The inclusion of the same population for all exposure and outcome data can help us obtain the most reliable results. That is what's actually done in many other literatures3, 4. However, from another perspective you are right, because the results obtained from the same population for all exposure and outcome data probably are inapplicable to other population. Thus, the relationship between anthropometric measures and sepsis revealed in this study need to be further demonstrated through including different population. This is undoubtedly a limitation of this study. Hence, we have supplemented this limitation in our discussion. Please see Page 16, Lines 320-324.

(Lines 320-324):

Firstly, only European populations were included. However, different cohorts of patients have different genetic profiles due to the difference of ethnic diversity, risk factors, and epigenetic profiles. Thus, the reliability of the causal relationships from our analysis should be demonstrated in other non-European descent.

6. In conclusion, your research addresses an important topic, and with some additional work, it could provide valuable insights into the study of sepsis risk. I look forward to your response and the subsequent version of your manuscript.

Our response: Thanks for your recognition.

Reviewer #2:

The authors performed a mendelian randomization (MR) study to investigate the link between anthropometric traits such as weight, BMI, height, etc., and sepsis and sepsis death. The manuscript lacks a detailed description of the methods used and plots that are normally included in MR studies (see comments below).

Our response: We appreciate the reviewer’s critical and constructive comments. It is our honour to get your help to revise our manuscript. Your comments give us valuable assistance in improving our manuscript. We have carefully revised our manuscript accordingly. After adopting your suggestions, we believe a significant improvement can be seen in our revised manuscript. The following are detailed responses to your recommendations.

1. - This is basically a one-sample MR since both the GWAS on the exposure and that on the outcome came from the same cohort (UK biobank). As this approach possess some limitations, these should be discussed in the text. Especially because two-samples MR studies on BMI and sepsis already exists.

Our response: We are very grateful for your review of our article carefully and for providing useful ideas. We agree with your view. Thus, we have supplemented this limitation in our discussion. Please see Page 16, Lines 320-324. On the other hand, there were indeed two-samples MR studies reported the relationship between BMI and sepsis. For example, Wang et al in 2023 indicated that their two-sample Mendelian randomization supports a causal relationship between BMI and sepsis, and thus further concluded that proper control of BMI may prevent sepsis5. However, BMI has received criticism recently for the following reasons6: 1) BMI falls short in its ability to distinguish between fat mass and non-fat mass (e.g., lean tissue mass), which might lead to the incorrect assessments of certain individuals. 2) BMI does not display the specific location of body fat which has been demonstrated to exert a vital role in forecasting the risk of several diseases. That’s why this study was designed to increase the understanding of the obesity-sepsis association by evaluating the causal effects of fat mass, nonfat mass and height on sepsis risk and mortality. 

(Lines 320-324):

Firstly, only European populations were included. However, different cohorts of patients have different genetic profiles due to the difference of ethnic diversity, risk factors, and epigenetic profiles. Thus, the reliabi

---

## [Decision Letter · Decision Letter 1]

24 Mar 2024

PONE-D-23-24676R1Mendelian randomization analysis reveals causal association of anthropometric measures on sepsis risk and mortality

PLOS ONE

Dear Dr. He,

Thank you for submitting your manuscript to PLOS ONE. After careful consideration, we feel that it has merit but does not fully meet PLOS ONE’s publication criteria as it currently stands. Therefore, we invite you to submit a revised version of the manuscript that addresses the points raised during the review process.

**While several concerns raised by previous reviewers have been appropriately addressed, it remains imperative to tend to the remaining points outlined by the reviewers. Specifically: i) enhance the description regarding the procedure for conducting multivariable MR; ii) revise the introduction to elucidate the methodology behind the selection of genetic IVs for the multivariable MR analysis; iii) integrate an overview of MR in the introduction, elucidating the three fundamental assumptions underlying MR analysis; iv) refine the introduction to minimize references to external publications and focus on elucidating the study's unique contributions, particularly concerning the investigation of body fat parameters.**

  Please submit your revised manuscript by May 08 2024 11:59PM. If you will need more time than this to complete your revisions, please reply to this message or contact the journal office at plosone@plos.org. Please include the following items when submitting your revised manuscript:A rebuttal letter that responds to each point raised by the academic editor and reviewer(s). You should upload this letter as a separate file labeled 'Response to Reviewers'.A marked-up copy of your manuscript that highlights changes made to the original version. You should upload this as a separate file labeled 'Revised Manuscript with Track Changes'.An unmarked version of your revised paper without tracked changes. You should upload this as a separate file labeled 'Manuscript'.

We look forward to receiving your revised manuscript.

Kind regards,

Giuseppe Remuzzi

Academic Editor

PLOS ONE

Reviewers' comments:

Reviewer's Responses to Questions

**Comments to the Author**

1. If the authors have adequately addressed your comments raised in a previous round of review and you feel that this manuscript is now acceptable for publication, you may indicate that here to bypass the “Comments to the Author” section, enter your conflict of interest statement in the “Confidential to Editor” section, and submit your "Accept" recommendation.

Reviewer #2: (No Response)

Reviewer #3: (No Response)

2. Is the manuscript technically sound, and do the data support the conclusions?

Reviewer #2: Partly

Reviewer #3: Yes

3. Has the statistical analysis been performed appropriately and rigorously? 

Reviewer #2: Yes

Reviewer #3: I Don't Know

4. Have the authors made all data underlying the findings in their manuscript fully available?

Reviewer #2: Yes

Reviewer #3: Yes

5. Is the manuscript presented in an intelligible fashion and written in standard English?

Reviewer #2: Yes

Reviewer #3: Yes

6. Review Comments to the Author

Reviewer #2: This study study is a one-sample MR not because the participants are predominantly European, but because there is a sample overlap between the GWASs on the exposures and the GWAS on sepsis due to the fact that they were performed on the same cohort (UK biobank). And this poses some limitations, for example on the use of MR-Egger (see here for example: https://doi.org/10.1093/ije/dyab084).

The procedure for multivariable MR needs to be better described, with particular attention to testing for weak instruments. You should also describe how genetic IVs were selected for the multivariable MR, for example in supplementary fig. 15 and supplementary table 3 each exposure has a different number of snps, while usually all snps associated with at least one exposure are included in the analysis before harmonization, LD based selection, weak instrument testing, etc.

- I would add a short description of MR, with the three assumptions, in the introduction.

- The following sentence might be rephrased: "Here, the least absolute shrinkage and selection operator (LASSO) regression was used to remove highly correlated exposures in the presence of various exposures"

- Line 127: "Exposures with significant adjusted p values of<0.05 were defined as exposures with potential evidence...". How did you adjust the p-values?

Reviewer #3: There seems to be a lot of repetition in the discussion on the effects of weight and height on sepsis and many portions seem redundant. Please avoid paraphrasing discussion section of another publication ( Wang et al, 2023). There is a very minimal mention of body fat parameters and specific discussion around those results. The sample overlap bias alluded to in the previous review has not been sufficiently addressed in the authors' responses especially if the authors state that this is a one sample MR. Why is two sample MR package being mentioned in the statistical section, it is a bit confusing. I think authors need to address this in a revision. While the results and analysis section is satisfactory, introduction and discussion needs to improve

7. PLOS authors have the option to publish the peer review history of their article (what does this mean?). If published, this will include your full peer review and any attached files.

Reviewer #2: **Yes: **Matteo Breno

Reviewer #3: No

---

## [Author Response · Author response to Decision Letter 1]

21 Apr 2024

[21 April 2024]

Editor-in-Chief

PLOS ONE

Dear Editor:

Thank you very much for your e-mail dated 17th April 2024, providing your requirement on our submission (Submission ID PONE-D-23-24676R2) entitled “Mendelian randomization analysis reveals causal association of anthropometric measures on sepsis risk and mortality”. We have removed funding-related text from the manuscript and would like to update our Funding Statement just as follows: 

"This work was supported by grants from the Natural Science Foundation of Fujian Province (2020J01227) and the Medical Innovation Science and Technology Project of Fujian Province (2020CXA047). The funder had vital role in study design, data collection and analysis, decision to publish, and preparation of the manuscript."

However, in online submission system, we could not found “Financial Disclosure section”, except “Financial information section”. Thus, we would like to update our Funding Statement via cover letter.

We thank the editor for your time. 

We are very grateful to you for your help in improving this article.

Sincerely,

Sincerely,

[Author’s name] 

He-fan He

[Affiliation]

Department of Anaesthesiology, the Second Affiliated Hospital of Fujian Medical University, Quanzhou, Fujian Province, China 

[Postal address]

No. 34 North Zhongshan Road, Quanzhou, Fujian Province, 362000, China. 

[Phone number] 

86 15683713870

[Email address]

1017118837@fjmu.edu.cn

---

## [Decision Letter · Decision Letter 2]

28 Jun 2024

PONE-D-23-24676R2Mendelian randomization analysis reveals causal association of anthropometric measures on sepsis risk and mortality

PLOS ONE

Dear Dr. He,

Thank you for submitting your manuscript to PLOS ONE. After careful consideration, we feel that it has merit but does not fully meet PLOS ONE’s publication criteria as it currently stands. Therefore, we invite you to submit a revised version of the manuscript that addresses the points raised during the review process.

****The authors have addressed most of the concerns raised by the reviewer. However, there are few minor comments that need to be addressed. Specifically: i) the issue with sample overlap is not addressed in the manuscript; ii) the reference cited in the response to reviewers was not added to the manuscript; iii) it should be included that MR is not immune to reverse **causation; iv) genetic associations should be included in supplementary table 1; v) t****he conclusion of the study needs better explanation.** 

We look forward to receiving your revised manuscript.

Kind regards,

Giuseppe Remuzzi

Academic Editor

PLOS ONE

Journal Requirements:

Reviewers' comments:

Reviewer's Responses to Questions

**Comments to the Author**

1. If the authors have adequately addressed your comments raised in a previous round of review and you feel that this manuscript is now acceptable for publication, you may indicate that here to bypass the “Comments to the Author” section, enter your conflict of interest statement in the “Confidential to Editor” section, and submit your "Accept" recommendation.

Reviewer #1: All comments have been addressed

Reviewer #2: (No Response)

2. Is the manuscript technically sound, and do the data support the conclusions?

Reviewer #1: Yes

Reviewer #2: Partly

3. Has the statistical analysis been performed appropriately and rigorously? 

Reviewer #1: Yes

Reviewer #2: Yes

4. Have the authors made all data underlying the findings in their manuscript fully available?

Reviewer #1: Yes

Reviewer #2: Yes

5. Is the manuscript presented in an intelligible fashion and written in standard English?

Reviewer #1: Yes

Reviewer #2: Yes

6. Review Comments to the Author

Reviewer #1: Thank you for the revisions made and for incorporating additional details on data analysis, such as sensitivity analysis. This manuscript effectively demonstrates a causal relationship between body mass and sepsis, providing valuable insights that could inform future clinical management strategies.

Reviewer #2: The sample overlap issue was not addressed in the manuscript, and the reference you cite in the response to reviewers was not added to the manuscript.

- Introduction, line 71. I suggest to move the MR assumptions here.

- Introduction, line 73. MR is not immune tho reverse causation, which can be tested by bidirectional MR.

- Introduction, line 75. With the right study design, you can test for causality with a regression (e.g., randomized controlled trials, lab experiment ).

- Methods, line 92: supplementary table 1 does not contain genetic associations

- Methods, line 105: see comment above

- Discussion: you start the discussione by saying (line 276-277): "...Our findings indicated that weight-related measures are closely related to sepsis risk and mortality..." and end it with "..but MVMR analysis indicated the observed effects for weight-related measures in the univariable MR analyses are more likely a bias caused by the interrelationship between anthropometric measures...". It is not easy to understand the conclusions that the authors are drawing.

7. PLOS authors have the option to publish the peer review history of their article (what does this mean?). If published, this will include your full peer review and any attached files.

Reviewer #1: No

Reviewer #2: **Yes: **Matteo Breno

---

## [Author Response · Author response to Decision Letter 2]

16 Jul 2024

Editor' Comments:

The authors have addressed most of the concerns raised by the reviewer. However, there are few minor comments that need to be addressed. 

Specifically: 

i) the issue with sample overlap is not addressed in the manuscript; 

ii) the reference cited in the response to reviewers was not added to the manuscript; 

iii) it should be included that MR is not immune to reverse causation; 

iv) genetic associations should be included in supplementary table 1; 

v) the conclusion of the study needs better explanation.

Our response: Thank you for taking the time to review our article. We have provided point-by-point responses to each of the reviewers’ questions. Please find them below. Thanks very much again! If there are any more problems, please do not hesitate to contact us. Best wishes!

Reviewers' Comments:

General response: We thank the reviewers for their time and insightful comments. We have carefully considered these comments and substantially revised the manuscript by thoroughly researching the field reviewed and clarifying the content of this manuscript to make it more comprehensive. Below, we address each of the reviewers’ questions.

Reviewer #1: Thank you for the revisions made and for incorporating additional details on data analysis, such as sensitivity analysis. This manuscript effectively demonstrates a causal relationship between body mass and sepsis, providing valuable insights that could inform future clinical management strategies.

Our response: We appreciate the reviewer’s critical and constructive comments. It is our honour to get your help to revise our manuscript. Thanks very much for your recognition. Best wishes!!

Reviewer #2: 

1.The sample overlap issue was not addressed in the manuscript, and the reference you cite in the response to reviewers was not added to the manuscript.

Our response: Thank you for your feedback and comments regarding the sample overlap issue. We are very grateful that the reviewer read our article carefully. As pointed previously, Pro. Minelli and coworkers(1) has demonstrated the feasibility of two sample MR package being used in one-sample MR analysis in 2021, thus after that some MR studies using the same cohort (such as UK biobank) were accepted for publication(2, 3). But it’s our negligence for omitting the important literature in our manuscript. We have added it in our revised manuscript. Please see Page 4, Lines 78-81. 

(Lines 78-81):

Importantly, Pro. Minelli and coworkers (1) further demonstrated the feasibility of two sample MR package being used in one-sample MR analysis in 2021, which promoting the application and development of MR in medical field(2, 3). 

2. Introduction, line 71. I suggest to move the MR assumptions here.

Our response: Thanks for your great suggestion, and we have moved the MR assumptions according to your advice. Please see Page 4, Lines 71-75. 

(Lines 71-75):

All of MR analysis were performed according to the following three key assumptions: (a) the multiple genetic variants included significantly involve in the exposure of interest; (b) no confounders affect the relationship between the exposure and outcome; (c) the genetic variants should influence the outcome only via their effect on exposure(4).

3. Introduction, line 73. MR is not immune the reverse causation, which can be tested by bidirectional MR.

Our response: We appreciate your good suggestion on improving the accuracy of the expression. We have deleted this incorrect expression. Please see Page 4, Lines 75-78. 

(Lines 75-78):

As a key epidemiological technique, MR analysis uses genetically instrumental variables (IVs) as proxies to explore the causality of exposure on outcome of interest, which overcomes these defects and bias caused by confounders, and other biases in observational studies(5).

4. Introduction, line 75. With the right study design, you can test for causality with a regression (e.g., randomized controlled trials, lab experiment).

Our response: Thanks very much for your constructive instruction. We have deleted this sentence. Your professional comments give us inspiration. Thanks very much! 

5. Methods, line 92: supplementary table 1 does not contain genetic associations

Our response: We appreciate the reviewer’s constructive comments. We have supplemented the genetic associations (including number of SNP, approximate variance explained, and mean F-statistic) in our supplementary table 1. Please see the following table:

6. Methods, line 105: see comment above

Our response: We have revised this according to your instruction above.

7. Discussion: you start the discussione by saying (line 276-277): "...Our findings indicated that weight-related measures are closely related to sepsis risk and mortality..." and end it with "..but MVMR analysis indicated the observed effects for weight-related measures in the univariable MR analyses are more likely a bias caused by the interrelationship between anthropometric measures...". It is not easy to understand the conclusions that the authors are drawing. 

Our response: We appreciate the reviewer’s critical comments. Just as pointed in our discussion, a new term “obesity paradox” is now popular in academic world. The reason is that high BMI is demonstrated to contribute to worse outcomes for critically ill patients in early works(6, 7), while a relationship with improved outcomes with an raised BMI was found in some more recent work(8-10). The underlying mechanism of this phenomenon remains unclear and understanding the genetic role of sepsis is perhaps the key to unravel the mysterious relationship between obesity and outcomes in patients with sepsis. Our results from univariable MR analysis supported the positive association of BMI with sepsis mortality at 28 days, which was in accordance with the outcomes from other literatures(11). However, these MR analyses failed to control for pleiotropic pathways resulted from the multicollinearity among anthropometric measures. Thus, we further performed MVMR analyses to assess the independent causal effects of BMI on sepsis risk utilizing lasso regression method. Our results demonstrated that the direct estimates for the BMI were significantly attenuated, causing a wider 95% CI that overlapped null. Hence, the observed effects for BMI in the univariable MR analyses are more likely a bias caused by the interrelationship between anthropometric measures. This probably explain the reason why “obesity paradox” occur in sepsis: namely previous literatures only paid attention to the association between BMI and sepsis and did not consider the collinear effect from other anthropometric measures. Thus, we concluded that these findings represent vital new knowledge on the role of anthropometric-related measures in the sepsis etiology.

Reference

1. Minelli C, Del Greco MF, van der Plaat DA, Bowden J, Sheehan NA, Thompson J. The use of two-sample methods for Mendelian randomization analyses on single large datasets. International journal of epidemiology. 2021;50(5):1651-9.

2. Lei P, Xu W, Wang C, Lin G, Yu S, Guo Y. Mendelian Randomization Analysis Reveals Causal Associations of Polyunsaturated Fatty Acids with Sepsis and Mortality Risk. Infectious diseases and therapy. 2023;12(7):1797-808.

3. Almramhi MM, Storm CS, Kia DA, Coneys R, Chhatwal BK, Wood NW. The role of body fat in multiple sclerosis susceptibility and severity: A Mendelian randomisation study. Multiple sclerosis (Houndmills, Basingstoke, England). 2022;28(11):1673-84.

4. Aad G, Abbott B, Abdallah J, Abdinov O, Aben R, Abolins M, et al. Search for Dark Matter in Events with Missing Transverse Momentum and a Higgs Boson Decaying to Two Photons in pp Collisions at sqrt[s]=8 TeV with the ATLAS Detector. Physical review letters. 2015;115(13):131801.

5. Haycock PC, Burgess S, Wade KH, Bowden J, Relton C, Davey Smith G. Best (but oft-forgotten) practices: the design, analysis, and interpretation of Mendelian randomization studies. The American journal of clinical nutrition. 2016;103(4):965-78.

6. Rattan R, Nasraway SA, Jr. Separating wheat from chaff: examining the obesity paradox in the critically ill. Critical care (London, England). 2013;17(4):168.

7. Nasraway SA, Jr., Albert M, Donnelly AM, Ruthazer R, Shikora SA, Saltzman E. Morbid obesity is an independent determinant of death among surgical critically ill patients. Crit Care Med. 2006;34(4):964-70; quiz 71.

8. Sakr Y, Alhussami I, Nanchal R, Wunderink RG, Pellis T, Wittebole X, et al. Being Overweight Is Associated With Greater Survival in ICU Patients: Results From the Intensive Care Over Nations Audit. Crit Care Med. 2015;43(12):2623-32.

9. Christopher KB. The Body Mass Index Paradox. Crit Care Med. 2015;43(12):2693-4.

10. Yeo HJ, Kim TH, Jang JH, Jeon K, Oh DK, Park MH, et al. Obesity Paradox and Functional Outcomes in Sepsis: A Multicenter Prospective Study. Crit Care Med. 2023;51(6):742-52.

11. Butler-Laporte G, Harroud A, Forgetta V, Richards JB. Elevated body mass index is associated with an increased risk of infectious disease admissions and mortality: a mendelian randomization study. Clinical microbiology and infection : the official publication of the European Society of Clinical Microbiology and Infectious Diseases. 2020.

---

## [Decision Letter · Decision Letter 3]

9 Sep 2024

Mendelian randomization analysis reveals causal association of anthropometric measures on sepsis risk and mortality

PONE-D-23-24676R3

Dear Dr. He,

We’re pleased to inform you that your manuscript has been judged scientifically suitable for publication and will be formally accepted for publication once it meets all outstanding technical requirements.

Kind regards,

Giuseppe Remuzzi

Academic Editor

PLOS ONE

Additional Editor Comments (optional):

**All the reviewers' issues have been appropriately resolved. Please ensure that the two minor changes requested by Reviewer 2 are made**

Reviewers' comments:

Reviewer's Responses to Questions

**Comments to the Author**

1. If the authors have adequately addressed your comments raised in a previous round of review and you feel that this manuscript is now acceptable for publication, you may indicate that here to bypass the “Comments to the Author” section, enter your conflict of interest statement in the “Confidential to Editor” section, and submit your "Accept" recommendation.

Reviewer #1: All comments have been addressed

Reviewer #2: All comments have been addressed

2. Is the manuscript technically sound, and do the data support the conclusions?

Reviewer #1: Yes

Reviewer #2: Partly

3. Has the statistical analysis been performed appropriately and rigorously? 

Reviewer #1: Yes

Reviewer #2: Yes

4. Have the authors made all data underlying the findings in their manuscript fully available?

Reviewer #1: Yes

Reviewer #2: Yes

5. Is the manuscript presented in an intelligible fashion and written in standard English?

Reviewer #1: Yes

Reviewer #2: Yes

6. Review Comments to the Author

Reviewer #1: (No Response)

Reviewer #2: I have just minor comments:

- Line 79, replace package with methods. The paper you cite is on methodology, not software packages.

- Line 367: remove causal. If the association seen by univariate MR is an artifact, as you state in the next sentence, the relation cannot be causal.

7. PLOS authors have the option to publish the peer review history of their article (what does this mean?). If published, this will include your full peer review and any attached files.

Reviewer #1: No

Reviewer #2: No

---

## [Editor Report · Acceptance letter]

20 Sep 2024

PONE-D-23-24676R3 

PLOS ONE

Dear Dr. He, 

I'm pleased to inform you that your manuscript has been deemed suitable for publication in PLOS ONE. Congratulations! Your manuscript is now being handed over to our production team.

Kind regards, 

on behalf of

Prof. Giuseppe Remuzzi 

Academic Editor

PLOS ONE